# MovingColor: Seamless Fusion of Fine-grained Video Color Enhancement

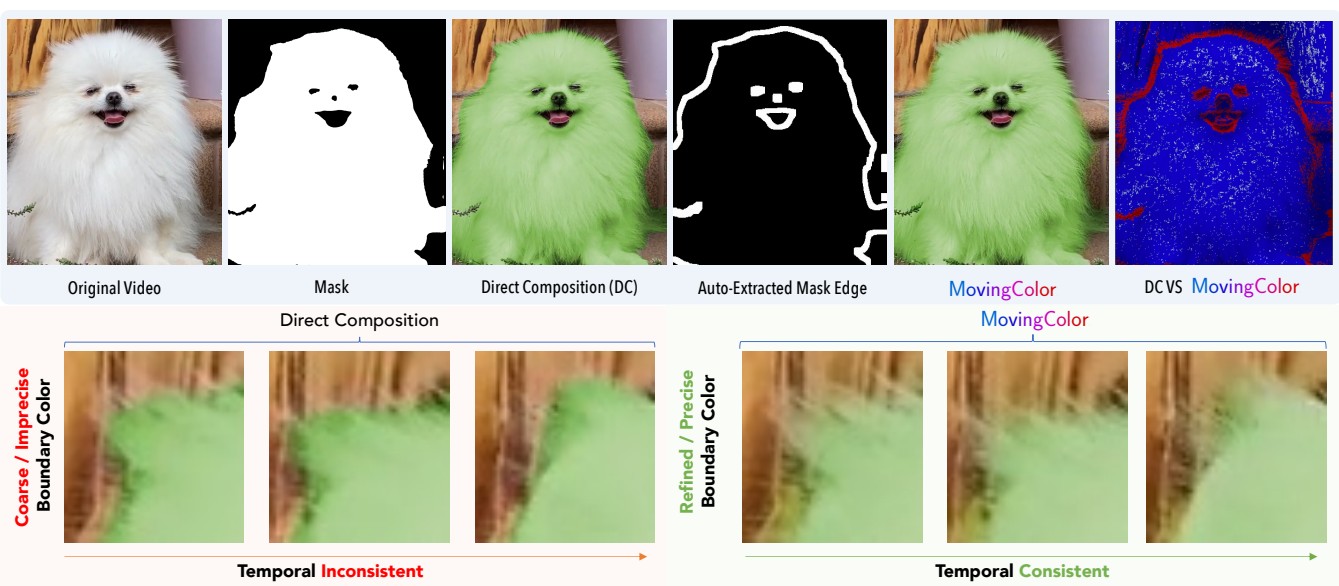

**Figure 1: MovingColor tackles the problem of natural fusion of fine-grained video color adjustments. It can achieve refined fusion of regional color adjustments and fuse the edit seamlessly into the input video and keep the temporal consistency in the meantime. (Best viewed in high-resolution color screen and please refer to supplementary video website (https://mm24-anonymous-id-279.github.io/) for full video and more examples to gain more accurate comparison.)**

## ABSTRACT

Fine-grained video color enhancement delivers superior visual results by making precise adjustments to specific areas of the frame, maintaining more natural color relationships compared to global enhancement techniques. However, dynamically applying these specific enhancements can lead to flickering artifacts and unsatisfactory color blending at object boundaries, issues caused by the coarse and unstable masks produced by current video segmentation algorithms. To overcome these challenges, we introduce MovingColor, featuring a novel self-supervised training approach that leverages large-scale video datasets. This approach redefines color fusion as a generation process using original full-frame textures and color editing information from non-edge areas. We address spatio-temporal inconsistencies with a spectral-spatial hybrid encoder that captures multi-scale spatial and frequency features, thus enhancing color adjustments in complex scenes. Additionally, our global-local feature propagation module, incorporating Transformer blocks, consolidates spatio-temporal contexts to ensure consistency among frames. Both quantitative and subjective evaluations validate the effectiveness of MovingColor in delivering state-of-the-art spatio-temporal consistency for video color enhancements, adhering closely to the intended color editing operations. These results demonstrate that MovingColor can effectively enhance fine-grained video color grading, making it more efficient and accessible to a wider range of users. We will release the code to support further research and practical applications.

## CCS CONCEPTS

• **Computing methodologies** → **Image processing**; *Computational photography*.

## KEYWORDS

Color Fusion, Video Color Enhancement, Video Editing

## 1 INTRODUCTION

Fine-grained video color enhancement enables visually appealing results by tailoring adjustments to specific regions to preserve accurate color relationships [31]. It has significant value in video editing, post-production, and creative applications. However, current fine-grained video color enhancement methods suffer from spatial and temporal inconsistencies due to limitations in video segmentation

and tracking techniques, resulting in imprecise and flickering masks that lead to spatial and temporal inconsistencies (Figure 1 and 2).

Existing approaches for video color enhancement, such as Harmonizer [14] and DeepLPF [24], attempt to address these challenges by adapting image-based techniques or employing localized filters. However, these methods struggle to maintain consistency across video frames and often fail to preserve the intended color edits. Additionally, video matting techniques like MODNet [15] are specialized for specific tasks and do not provide generalizable solutions for color fusion.

To address these issues, we propose MovingColor, a novel self-supervised learning method for video color fusion that effectively ensures both spatial and temporal consistency. MovingColor formulates color fusion as generating natural fusion results given the original full texture information, color editing information in the non-edge area, and the mask edge, leveraging vast amounts of unlabeled video data. MovingColor employs a spectral-spatial hybrid encoder, combining convolutional and Fast Fourier Convolutional networks to capture multi-scale spatial features and frequency information. A global-local feature propagation module with Transformer blocks aggregates spatio-temporal contexts across frames for consistency. MovingColor captures meaningful spatial and temporal features from the video data, encoding both global and local information across frames to enable consistent color fusion.

To comprehensively evaluate color fusion performance, we introduce the D5 dataset, synthesized using 3D software. It features diverse 4K video clips from various scenes, with ground truth mattings for all objects. Extensive experiments demonstrate that MovingColor achieves state-of-the-art results, outperforming existing color manipulation, harmonization, and video consistency methods on the D5, DAVIS [26], and YouTube-VOS [35] datasets. Furthermore, MovingColor exhibits robustness to varying edge ratios, resolutions, and color adjustments, highlighting its adaptability to diverse real-world scenarios.

The main contributions are: 1) A novel self-supervised learning approach for spatially and temporally consistent video color fusion; 2) The design of MovingColor with the spectral-spatial hybrid encoder and the global-local feature propagation module for capturing multi-scale features and aggregating spatio-temporal contexts; 3) The D5 dataset for comprehensive evaluation of color fusion performance; and 4) State-of-the-art results on 3 datasets, outperforming existing methods in quantitative metrics and user studies.

## 2 RELATED WORKS

### 2.1 Fine-grained Video Color Enhancement

Video color enhancement aims to improve the aesthetic appeal of videos by adjusting colors. Techniques such as Harmonizer [14] and PSENet [33], initially developed for image enhancement, are adapted for video by ensuring color consistency within shots. Similarly, methods using global 3D-LUTs, like 3DLUT [39] and AdaInt [38], also maintain consistent LUT settings across clips. However, they lack the ability to make detailed, region-specific color adjustments.

Traditional tools such as Adobe After Effects and DaVinci Resolve, which allow detailed regional color modifications, inspire spatial mask-based techniques like DeepLPF [24], DCCF [36], and LED-Net [42]. These methods employ localized filters for precise color

grading but struggle with maintaining consistency across video frames. RSFNet [25] attempts to address this by using segmentation-based masks (e.g., Mask2Former [5], SegFormer [1], SAM [17]) for color enhancement. Nonetheless, these masks often fail to ensure smooth color transitions, particularly during substantial edits, leading to performance issues.

Our approach, MovingColor, targets these challenges by improving both spatial and temporal color consistency, especially focusing on issues related to the instability of segmentation masks.

### 2.2 Spatial Inconsistency

Spatial inconsistency in video color editing involves refining masks for precise segmentation and employing post-processing techniques like harmonization and color matching to preserve original edits and ensure uniformity.

Mask refinement methods like HQ-SAM [13] and Seg-Refiner [32] improve the accuracy of masks for finely detailed objects but often struggle with creating natural transitions at complex edges. In image harmonization, approaches such as Harmonizer [14], S2CRNet [21], and DCCF [37] effectively merge foreground and background but may alter intended color edits and are less effective in video due to mask inconsistencies.

Style transfer and inpainting techniques, although they adjust an image's style or fill in missing areas, frequently alter colors and textures inappropriately for video tasks. Techniques like ReCoRo [34] focus on localized lightness adjustments but are limited by their exclusive focus on this aspect and reliance on specific datasets.

Our proposed solution effectively manages these challenges, even with the coarse and unstable masks from other tracking techniques, ensuring smooth color transitions and consistency.

### 2.3 Temporal Inconsistency

Temporal inconsistencies arise when image processing algorithms are directly applied to videos frame-by-frame. Existing methods like Blind Video Temporal Consistency [2] and Learning Blind Video Temporal Consistency [18] aim to reduce these discrepancies. While they enhance temporal stability, they often degrade the video's quality and diminish the efficacy of the original processing methods.

Video style transfer methods attempt to maintain visual consistency over time by aligning frames with a reference style, as seen in Stylizing Video [12] and Interactive Video Stylization [30]. However, they generally fall short in maintaining user edits and have limited applicability to real-world photography.

MovingColor, our proposed method, effectively minimizes temporal inconsistencies while preserving the desired color qualities of the edited areas, suitable for various contexts including artistic and photographic enhancement.

## 3 COLOR FUSION AND SELF-SUPERVISED LEARNING SCHEME

### 3.1 Color Fusion and Challenges

As demonstrated in Figure 2a, fine-grained video color enhancement yields visually superior results by tailoring adjustments to specific frame regions, thus preserving natural color relationships

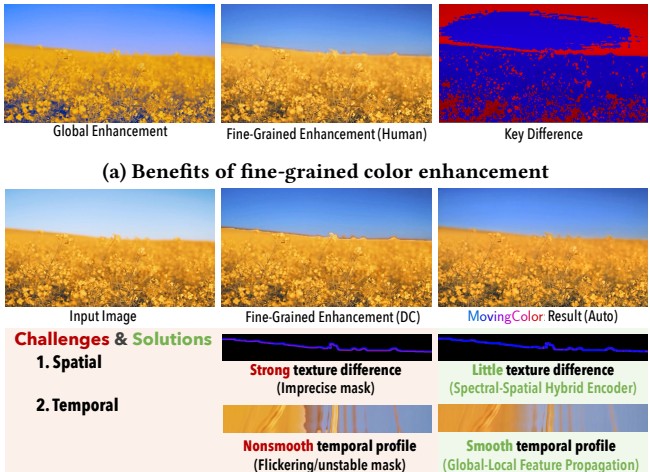

(a) Benefits of fine-grained color enhancement

(b) Challenges in color fusion and solutions in MovingColor

**Figure 2: Fine-grained color enhancement is crucial for maintaining accurate color relationships, but it often introduces inconsistencies. MovingColor effectively resolves these issues in natural fusion of fine-grained color edits.**

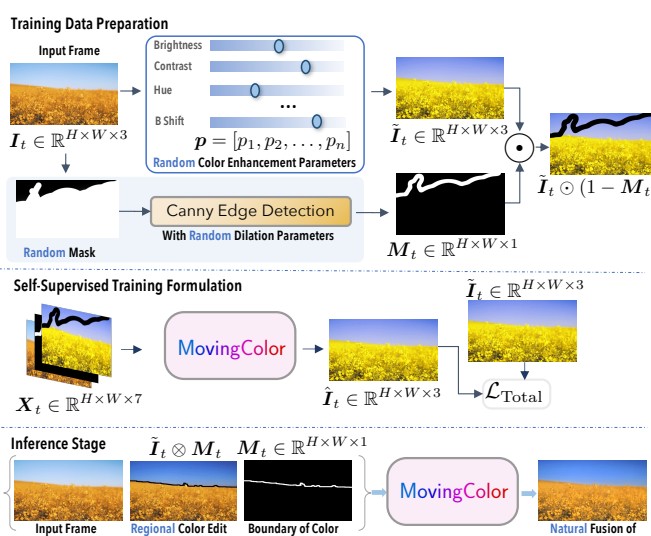

**Figure 3: A novel self-supervised training methodology for color fusion in video data. (For easy of visualization, we show the input and output of the network for a single frame. The network is trained on a sequence of frames as detailed in the following sections.)**

better than global enhancement methods, which can introduce unnatural hues such as blue in fields or pink in skies. However, as Figure 2b highlights, this approach introduces challenges related to mask precision. Current video matting technologies do not yet provide accurate tracking, leading to spatial inconsistencies from imprecise masks—evident from stark texture variations where high-frequency image components are improperly altered—and temporal inconsistencies from flickering masks, as shown by the non-smooth temporal profiles created by stacking horizontal pixel rows from consecutive frames. Our method, MovingColor, addresses these issues effectively, ensuring both spatial and temporal consistency as evidenced in the figures.

### 3.2 Gaps in Video Matting and Edge Refinement

For a given input frame $I_t$, color transformation $T$, and binary mask $A_t$, our objective is to achieve realistic color fusion. Typically, existing methods compute a blending matte $B_t$ as follows:

$$O_t = A_t \odot T(I_t) + (1 - A_t) \odot B_t \qquad (1)$$

However, obtaining accurate $B_t$ for video sequences is challenging, as video matting datasets primarily include only foreground elements without the corresponding $(I_t, B_t)$ pairs. Consequently, methods are often trained on synthetic datasets, which may not perform well in practical applications. Current video matting solutions are typically specialized, such as portrait video matting [15], and do not provide generalizable solutions. Consequently, colorists frequently resort to semi-automatic tools like RotoBrush V3 in Adobe After Effects, which require extensive manual adjustments and are both labor-intensive and time-consuming.

### 3.3 Self-Supervised Problem Formulation

*3.3.1 Problem Formulation.* MovingColor introduces a novel self-supervised learning approach for video color fusion that leverages

vast amounts of readily available unlabeled video data. Unlike supervised learning methods that rely on labeled data, MovingColor formulates color fusion as a pretext task of drawing conditional pixel samples from a learned distribution $p_\theta(\tilde{I}_t | I_t, T(I_t) \odot (1 - M_t), M_t)$ parameterized by $\theta$. This task encourages the model to learn useful representations for fusing color edits based on full texture information in original video frame $I_t$, color editing information in non-edge area $\tilde{I}_t \odot (1 - M_t)$, and the mask edge $M_t$. By solving this pretext task, MovingColor learns to capture meaningful spatial and temporal features and representations from the video data, encoding both global and local information across frames to enable spatially and temporally consistent color fusion. The learned representations capture the inherent structure and patterns within the video frames, allowing the model to perform seamless color blending without requiring human labeling.

*3.3.2 Training Data Preparation.* To enhance the robustness of the pretext task learning, we introduce randomness in color adjustment and mask generation, leveraging the diverse content and color distributions in the large-scale dataset. A random parametric Look-Up Table (LUT) generator, parameterized by $p$, modifies video frame batches to create color-enhanced versions $\tilde{I}_t$. Randomly generated masks $M_t$, incorporating both static and dynamic elements, are obtained using approaches similar to [16]. The training input $X_t = \text{Concate}[I_t, \tilde{I}_t \odot M_t, M_t]$ is formed in $\mathbb{R}^{H \times W \times 7}$, with $\tilde{I}_t$ serving as the ground truth.

*3.3.3 Inference Phase.* During the inference stage for video color enhancement, the system processes input frames, masks, and color-adjusted frames, addressing spatio-temporal inconsistency at mask edges, such as where the sky meets the field, as shown in the figure 3. Inference is performed in a sliding window fashion over

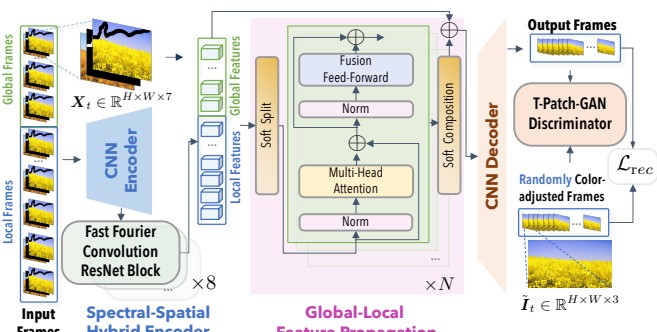

**Figure 4: Architecture of MovingColor, showcasing a hybrid encoder combining convolutional and Fast Fourier networks for multi-scale feature extraction, complemented by a Transformer-based global-local propagation module to enhance spatio-temporal consistency.**

batches of frames. Local neighboring frames defined by the window size and global frames spaced by a stride are selected. The model performs feature propagation and transformer-based video color fusion, outputting the predicted frames. Results from overlapping windows are averaged for the final output.

## 4 METHOD

### 4.1 Architecture Overview

As shown in Figure 4, the proposed MovingColor employs a spectral-spatial hybrid encoder with convolutional and Fast Fourier Convolutional networks to capture multi-scale spatial features and frequency information, followed by a global-local feature propagation module with Transformer blocks to aggregate spatio-temporal contexts across frames for spatial and temporal consistency. Due to space constraints, we briefly introduce the key components here, with more comprehensive details available in the supplementary materials. Additionally, we will make our code available for reproduction and application purposes.

### 4.2 Spectral-Spatial Hybrid Encoder

Seeking to combine the benefits of both convolutional and Fourier techniques [6], we propose chaining a convolutional encoder with a Fast Fourier Convolutional encoder into one hybrid architecture. This spectral-spatial design allows efficient learning of multi-scale spatial features while also capturing frequency information, providing a rich yet compact input representation for color fusion.

*4.2.1 Convolutional Encoder.* We use a lightweight encoder that is the same as that in FuseFormer [23], which is adapted from STTN [40] with deeper layers, more channel growth in the same time, using the group convolutions in the later layers to limit model complexity. Features from earlier layers are concatenated to later ones using grouped convolutions to aggregate multi-scale representations. The encoder outputs a compact yet rich representation of the input for further processing.

*4.2.2 Fast Fourier Convolution Encoder.* The Fast Fourier Convolution (FFC) Encoder consists of 8 chained FFC blocks that learn hierarchical features. Each FFC Block contains two FFC layers with

split local and global streams. Specifically, both local and global streams are allocated half of the intermediate channels. The FFC convolutions use a kernel size of 3, ReLU activation, and batch normalization applied separately to both streams after convolution. Local-to-local and global-to-global components use spectral Fourier transforms while cross-stream transforms are spatial convolutions. Each block takes the output of the previous block as input, combining it with the FFC convolution outputs using residual connections. In summary, stacking 8 FFC blocks forms a flexible FFC encoder that jointly learns hierarchical spectral-spatio, local and global representations.

### 4.3 Global-Local Feature Propagation

The module for global-local frame feature propagation is designed to improve temporal consistency. It is architecturally composed of $N$ temporal Transformer blocks.

*4.3.1 Temporal Transformer Block.* Each temporal transformer block undergoes a series of transformations, beginning with layer normalization applied to input token features. This step stabilizes the learning process and normalizes feature distributions. The normalized features are then processed through the Self-Attention module for aggregating spatio-temporal contextual information. Following the attention mechanism, we integrate a residual connection, allowing the addition of the attention outputs to the original input token features. This approach ensures the retention of critical information through the network layers. Another layer normalization is then applied to these combined features, setting the stage for the subsequent Fusion Feed-Forward Network [23]. This network, with its series of linear transformations and GELU activations, further refines the representations, endowing the model with the capacity to capture complex, high-level abstractions from the data.

*4.3.2 Global-Local Frame Feature Propagation.* Reflecting insights from prior studies [8, 20], the exclusive reliance on local temporal neighbors is acknowledged to be informationally constraining. Incorporating global-level information can enhance the synthesis of naturalistic color and textures. To this end, non-local frames are integrated to introduce a broader context. The process involves a soft split operation for embedding overlapping patches from both local and non-local temporal features. At the end of the feature propagation module, we use a soft composite operator to composite the embedded tokens to features.

This architecture, encompassing temporal Transformer blocks along with global-local frame features, is adept at forging a detailed and layered representation of spatio-temporal dynamics. It aggregates features across spatial and temporal scales effectively. The resulting token features offer a holistic view of the input video data, capturing the nuanced interplay of spatial and temporal elements.

### 4.4 Training and Loss function

In each iteration, the model processes input frames, masked out frames, and masks. The generator, which encompasses feature encoding, propagation and decoding is responsible for predicting the output frames. The model utilizes a combination of loss functions to optimize video frame generation.

**Reconstruction Loss**: To measure the discrepancy between the output video sequence $\hat{Y}$ and the ground-truth sequence $Y$, an L1 loss is employed, defined as $\mathcal{L}_{\text{rec}} = \left\| \hat{I}_t - \tilde{I}_t \right\|_1$.

**Adversarial Loss**: Furthermore, we introduce an adversarial training procedure with a T-PatchGAN based discriminator $D$ [3] to enhance the quality and coherence of generated videos. The discriminator's loss function, $\mathcal{L}_D$, optimizes it to differentiate between real and generated frames and is defined as $\mathcal{L}_D = \mathbb{E}_{\tilde{I}_t}[\log D(\tilde{I}_t)] + \mathbb{E}_{\hat{I}_t}[1 - \log D(\hat{I}_t)]$. For the generator, the adversarial loss is calculated as $\mathcal{L}_G = -\mathbb{E}_{\hat{I}_t}[\log D(\hat{I}_t)]$.

**Perceptual Loss**: The perceptual loss $\mathcal{L}_{\text{perc}}$, calculated between the VGG-19 [27] feature maps of the output and ground truth, facilitates similarity assessments within the semantic feature space.

**Total Loss**: The total loss is a weighted sum of these components: $\mathcal{L}_{\text{total}} = \mathcal{L}_{\text{rec}} + \lambda_1 \mathcal{L}_G + \lambda_2 \mathcal{L}_{\text{perc}}$, with the weight $\lambda_1 = 0.01$, $\lambda_2 = 0.5$.

# 5 EXPERIMENTS

## 5.1 Datasets

*5.1.1 DAVIS and YouTube-VOS.* DAVIS [26] and YouTube-VOS [35] are two widely used video object segmentation datasets. DAVIS contains 50 video sequences with pixel-level annotations. YouTube-VOS is a large-scale dataset with 3,471 video sequences. We use the training set of YouTube-VOS to train MovingColor. The reference-based similarity metrics for the DAVIS and YouTube-VOS datasets are calculated exclusively in non-edge regions due to the absence of ground truth for evaluating color fusion on edge areas, as these datasets provide only coarse, unrefined masks.

*5.1.2 D5 Dataset.* We propose the D5 dataset to address the limitations of existing video object segmentation datasets, which lack precise masks for evaluation, and video matting datasets that only contain foreground object mattings. Synthesizing results by compositing foreground mattings with background videos often yields unnatural outputs that do not accurately reflect lighting and color interactions. Using D5 Render, a 3D software, we generate 121 diverse 4K video clips from 12 scenes, featuring various camera movements and moving subjects such as people, animals, and architecture. Each video includes material ID maps serving as ground truth masks for all objects. This dataset is suitable for evaluating color fusion and related tasks, *e.g.* matting and composition.

## 5.2 Experimental Settings

*5.2.1 Implementation Details.* We use the training set of YouTube-VOS [35] with 3471 video sequences for training and did not fine-tuned on other datasets. We use the Adam optimizer with a learning rate of 0.0001. The batch size is set to 4. The training process takes about 2 days on 4 NVIDIA V100 GPUs. For more implementation settings, please refer to the supplementary materials.

*5.2.2 Baselines.* To the best of our knowledge, we are the first to introduce a color fusion method, and thus lack direct comparative methods in this area. Instead, we compare our approach with related color manipulation techniques: the color matching method Color Matcher [11], image harmonization methods (Harmonizer [14], S2CRNet [22], and PCTNet [9]), and the style transfer method

StyA2K [43] and FSPBT [30]. We also compare with the video consistency method All-In-One-Deflicker [19].

*5.2.3 Evaluation Metrics.* We employ quantitative metrics to evaluate video color fusion performance, measuring reference-based similarity, texture preservation, temporal consistency, and efficiency.

**Reference-based Difference**: We use PSNR, SSIM, ΔE, and the perceptually-aligned DreamSim [7] between result and ground truth frames. Higher PSNR, SSIM, and DreamSim values and lower ΔE indicate better similarity and color accuracy. For the D5 dataset, we apply these metrics to full frames, while for DAVIS and YouTube-VOS, we compare only non-edge areas due to the lack of ground truth for edge regions.

**Texture Preservation**: We introduce the texture difference (TD) metric to assess the preservation of texture details, inspired by digital art practices [4] and image processing workflows [16]. TD measures the difference in high-frequency texture components between result and input frames, with lower values indicating better preservation without unwanted textures.

**Temporal Consistency**: We employ flow warping error ($E_{\text{Warp}}$) [18], Patch Consistency (PC) [10], and Perceptual Video Clip Similarity (PVCS) [29] to assess visual steadiness across frames. Lower PVCS and $E_{\text{Warp}}$ values and higher PC scores indicate better temporal coherence. For brevity, we report $E_{\text{Warp}} \times 10^3$, denoted as $E^*_{\text{Warp}}$.

**Efficiency**: We measure FLOPs and inference time per frame.

## 5.3 Quantitative Evaluation

As shown in Table 1, MovingColor achieves the best performance on the D5-Material dataset in terms of reference-based similarity, with the highest PSNR (26.88), SSIM (0.88), and DreamSim (0.10) scores, and the lowest ΔE (4.35). For texture preservation, MovingColor and Color Matcher both achieve the lowest TD (1.03), demonstrating their effectiveness in maintaining texture details without introducing artifacts. In terms of temporal consistency, MovingColor performs competitively, with the highest $PC_{\text{SSIM}}$ (0.97) and the second-lowest PVCS (0.54) and $E^*_{\text{Warp}}$ (0.29) scores.

MovingColor maintains a good balance between performance and efficiency, with a runtime of 0.08s per frame, which is comparable to the fastest methods (Harmonizer and S2CRNet) while achieving significantly better results. Although MovingColor has higher FLOPs (26.51G) compared to some baselines, it is still more efficient than FSPBT and the combination of StyA2K and Deflicker. These results demonstrate that MovingColor achieves state-of-the-art performance in video color fusion, successfully balancing reference-based similarity, texture preservation, temporal consistency, and efficiency across multiple datasets and evaluation metrics. Additional results on DAVIS and YouTube datasets are provided in the supplementary materials.

*5.3.1 Comparison with Adapted Inpainting Methods.* Figure 5 shows that simply adapting existing inpainting methods to the color fusion setting yields unsatisfactory results. We compare MovingColor to LAMA [28] and Propainter [41], both adapted to take the same input as our method. Since LAMA is an image inpainting method that performs poorly in terms of temporal consistency, we add the same temporal module as MovingColor for a fair comparison. Despite this modification, MovingColor significantly outperforms

**Table 1: Color fusion performance comparisons between Color Matcher, Harmonizer, S2CRNet, StyA2K, PCTNet, Deflicker, StyA2K+Deflicker and our method on D5-Material dataset. Additional results on the DAVIS and YouTube datasets, exhibiting similar trends, are provided in the supplementary materials.**

| Category | Method | Reference-based Difference | | | | Texture | Temporal Consistency | | | | Efficiency | |
|---|---|---|---|---|---|---|---|---|---|---|---|---|
| | | PSNR↑ | SSIM↑ | DreamSim↓ | ΔE↓ | TD↓ | $PC_{PSNR}$ ↑ | $PC_{SSIM}$ ↑ | PVCS↓ | $E^*_{Warp}$ ↓ | FLOPs↓ | RunTime↓ |
| Space | Color Matcher | 21.74 | 0.81 | 0.11 | 11.11 | **1.03** | 38.63 | 0.96 | 0.62 | 0.31 | - | 0.03 |
| | Harmonizer | 20.19 | 0.72 | 0.12 | 12.54 | 1.20 | 35.76 | 0.95 | 0.58 | 0.64 | **3.60M** | **0.01** |
| | S2CRNet | 21.32 | 0.80 | 0.13 | 10.41 | 1.75 | 33.99 | 0.94 | 0.75 | 0.92 | 0.10G | 0.02 |
| | PCTNet | 23.42 | 0.75 | 0.12 | 8.91 | 1.47 | 35.35 | 0.95 | **0.50** | 0.67 | 1.30G | 0.02 |
| | StyA2K | 20.58 | 0.78 | 0.14 | 9.83 | 1.17 | 35.85 | 0.96 | 0.82 | 0.68 | 10.21G | 0.04 |
| Time | Deflicker | 16.57 | 0.41 | 0.12 | 12.92 | 2.18 | 38.45 | 0.96 | 1.61 | 0.44 | 998.03G | 4.72 |
| Space + Time | FSPBT | 23.55 | 0.85 | 0.14 | 5.56 | 1.24 | **39.86** | **0.97** | 0.71 | **0.25** | 122.45G | 1.55 |
| | StyA2K+Deflicker | 15.07 | 0.38 | 0.15 | 17.39 | 1.93 | 38.25 | **0.97** | 1.68 | 0.45 | 1008.24G | 5.11 |
| **Ours** | **MovingColor** | **26.88** | **0.88** | **0.10** | **4.35** | **1.03** | 38.61 | **0.97** | 0.54 | 0.29 | 26.51G | 0.08 |

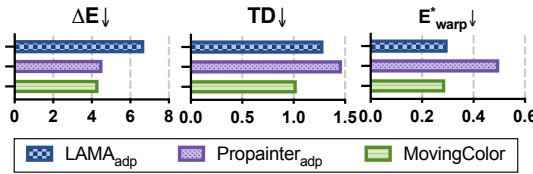

**Figure 5: Comparison with adapted inpainting methods on D5 dataset. Simple adaptation of existing efforts cannot achieve both spatial and temporal consistency and harmony.**

both LAMA and Propainter on all datasets. Due to space constraints, we present results on the D5 dataset; however, similar trends are observed on other datasets. These comparisons, along with the ablation studies, further validate the effectiveness of MovingColor's design choices for color fusion tasks.

## 5.4 Ablation Studies

We conduct ablation studies to investigate the effectiveness of key components in MovingColor, including the Fast Fourier Convolution (FFC) encoder and the global-local feature propagation. As shown in Table 2, the most substantial improvement is observed when integrating the FFC encoder along with both local and global features, achieving the lowest ΔE, TD, and $E^*_{Warp}$ scores across all datasets (D5, DAVIS, and YouTube-VOS). Comparing the variants with and without the FFC encoder, we observe a significant reduction in ΔE when FFC is included, highlighting the importance of capturing spectral information for accurate color fusion. Similarly, the inclusion of global features consistently improves performance across all metrics and datasets, underscoring the benefits of incorporating non-local temporal information. These ablation results conclusively demonstrate the efficacy of the FFC encoder and the synergistic effect of global-local feature propagation in enhancing MovingColor's performance. In addition, the effectiveness of the loss functions has been validated, with detailed loss ablation studies provided in the supplementary material.

## 5.5 Robustness Evaluation

To evaluate MovingColor's adaptability and consistency, we perform an extensive robustness test, considering various edge ratios,

**Table 2: Comparative results of structural variants. F stands for FFC, L stands for local frames, and G for global frames.**

| Var | | | D5 | | | DAVIS | | | YouTube-VOS | | |
|---|---|---|---|---|---|---|---|---|---|---|---|
| F | L | G | ΔE ↓ | TD↓ | $E^*_{Warp}$ ↓ | ΔE ↓ | TD↓ | $E^*_{Warp}$ ↓ | ΔE ↓ | TD↓ | $E^*_{Warp}$ ↓ |
| | ✓ | ✓ | 7.65 | 1.47 | 0.50 | 6.88 | 2.11 | 1.33 | 8.28 | 1.25 | 0.76 |
| ✓ | | ✓ | **3.39** | 1.04 | 0.30 | 2.93 | 2.04 | 1.08 | **3.14** | 1.13 | 0.52 |
| ✓ | ✓ | | **3.39** | 1.06 | 0.33 | 2.94 | 2.09 | 1.12 | **3.14** | 1.17 | 0.55 |
| ✓ | ✓ | ✓ | **3.39** | **1.03** | **0.29** | **2.93** | **2.02** | **1.06** | **3.14** | **1.10** | **0.51** |

input resolutions, and color adjustments. Figure 6 showcases the results, demonstrating the model's capability to deliver high-quality output across diverse and demanding conditions.

### 5.5.1 Robustness to Varying Edge Ratios.
Figure 6a evaluates MovingColor's robustness to edge ratios ranging from 5% to 25% on the D5, DAVIS, and YouTube datasets, covering typical real-world scenarios. Across all datasets, ΔE increases with larger edge ratios due to less available color information for fusion, while texture difference (TD) improves as larger fusion areas allow for smoother transitions and more natural textures.

On D5 and YouTube, MovingColor maintains stable performance, with slight increases in mean ΔE (3.09 to 3.55) and consistent mean TD (around 1.1). DAVIS, containing fast-moving objects and more occlusions, presents a more challenging scenario. Still, MovingColor outperforms all baselines and maintains stable performance, with mean ΔE rising from 2.8 to 3.48 and mean TD improving from 2.06 to 1.81 as edge ratio increases, demonstrating its effectiveness in preserving original color adjustments and textures for better spatial consistency and harmony.

### 5.5.2 Robustness to Varying Resolutions.
Figure 6b evaluates MovingColor's performance across different resolutions on the DAVIS and YouTube datasets. As resolution increases, both ΔE and TD exhibit consistent improvements, indicating better color accuracy and spatial consistency at higher resolutions.

### 5.5.3 Robustness to Different Color Adjustments.
Figure 6c demonstrates MovingColor's robustness to various color adjustments on the D5 dataset, with results for other datasets provided in the supplementary material. We evaluate the impact of increasing (+) and

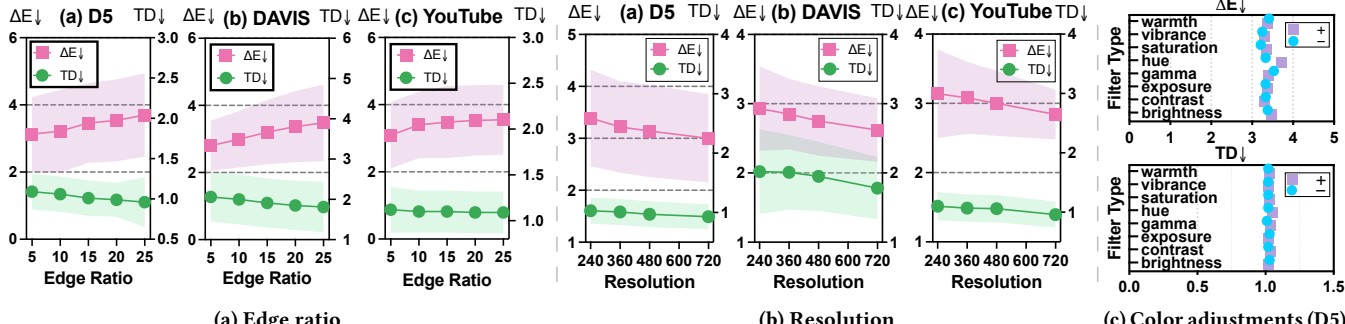

**Figure 6: Robustness evaluation on varying edge ratio, resolution and color adjustments.**

decreasing (-) color adjustment parameters such as brightness, contrast, exposure, gamma, hue, saturation, vibrance, and warmth on the ΔE and TD metrics. Across different color adjustments, ΔE remains stable, with most values ranging from 3.2 to 3.5. The highest ΔE occurs when increasing hue (3.71), while the lowest is observed when decreasing saturation (3.21), indicating consistent color accuracy regardless of the specific adjustment. Similarly, TD values remain close to 1.0 for all adjustments, highlighting stable color blending performance. These results showcase MovingColor's robustness to color adjustments, delivering consistent color fusion performance across diverse scenarios.

## 5.6 User Study

Evaluation metrics based solely on reference-based similarities have limitations, as there can be multiple reasonable results for color fusion tasks. Therefore, we conducted a user study with 68 participants. The supplementary material contains comprehensive information on the user study settings and an interview with a professional colorist. The baselines we selected cover key capabilities needed for video color fusion: handling spatial inconsistencies (Color Matcher, StyA2K), temporal inconsistencies (Deflicker), and combinations thereof (Deflicker+StyA2K). For the study, we cropped areas with noticeable artifacts from 3 videos to better showcase details.

As shown in Figure 7, the user study results demonstrate MovingColor's superior performance over the baselines across three aspects: spatial consistency, temporal consistency, and color accuracy. Specifically, MovingColor substantially outperforms the baselines in enhancing spatial consistency, with Color Matcher being a reasonably strong baseline by maintaining color harmony. Although StyA2K can perform good style transfers, it still suffers from unrealistic color transitions and inconsistencies. Deflicker alone does little to improve spatial consistency. The Deflicker+StyA2K combination, while combining the strengths of both methods, still falls short of MovingColor. Regarding temporal consistency, MovingColor performs comparably to the strong ColorMatcher baseline and better than StyA2K and Deflicker. This shows MovingColor's ability to maintain consistent colors over time. Finally, MovingColor has the lowest mean ΔE score, indicating it most accurately preserves target colors with minimal deviation. Together, these user study findings demonstrate MovingColor's state-of-the-art performance in video color fusion across key criteria.

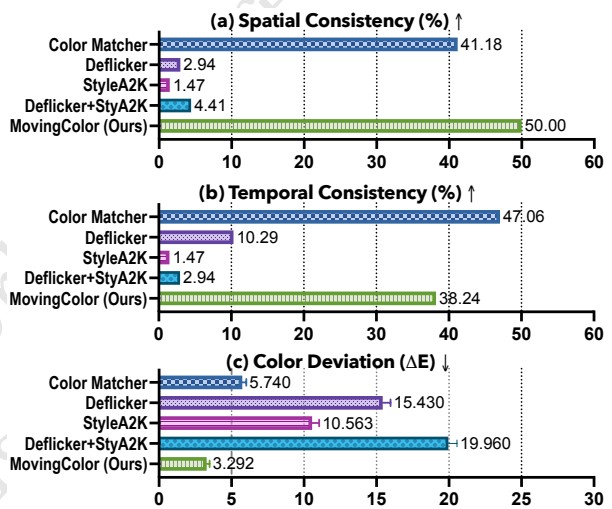

**Figure 7: User study results for (a) spatial consistency, (b) temporal consistency, and (c) color deviation (ΔE). Moving-Color achieves the best spatial consistency and comparable temporal consistency to the statistical method ColorMatcher, while maintaining the lowest color deviation.**

## 5.7 Visual Results

Figure 8 demonstrates MovingColor's superior color fusion performance compared to state-of-the-art methods. It effectively maintains spatio-temporal consistency while preserving intended colors in non-edge regions, outperforming baselines. Visit https://mm24-anonymous-id-279.github.io/ for more video results.

## 6 LIMITATIONS AND FUTURE WORK

MovingColor struggles to accurately process long, thin objects, such as sticks, and transparent objects. This limitation stems from the difficulty in segmenting long, thin objects and transparent objects, which remains a challenging problem in computer vision. Addressing this issue and improving the method's ability to handle these complex objects will be a focus of our future work. Despite their limitations in generating high-resolution, long videos, Diffusion-based models show promise for enhanced color fusion, which we aim to explore in the future.

**Figure 8: Example visual results. The results show that MovingColor is the only method that can achieve both spatial temporal consistency while with minimal color deviation in the non-edge area. Please refer to the supplementary website (https://mm24-anonymous-id-279.github.io/) for more video results.**

## 7 CONCLUSION

We introduce MovingColor, a novel self-supervised learning approach designed for seamless fusion of fine-grained video color enhancement, ensuring both spatial and temporal consistency. By employing a hybrid encoder and feature propagation mechanisms, MovingColor effectively addresses the challenges of color inconsistencies across frames. Extensive experiments demonstrate that MovingColor outperforms state-of-the-art methods, achieving superior color accuracy and consistency while exhibiting robustness to diverse scenarios. This work paves the way for efficient, high-quality fine-grained video color enhancement, with potential implications for both academic research and industrial applications. The code will be made publicly available to facilitate future research and real-world applications.

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
