# OpenReview forum: "MovingColor: Seamless Fusion of Fine-grained Video Color Enhancement"
_acmmm.org/ACMMM/2024/Conference — MM2024 Oral_

### Official Review · Reviewer_PpfK · 2024-05-12

**Rating:** 4
**Confidence:** 3

**Summary:**

The paper introduces "MovingColor," a post-processing method for fine-grained color enhancement in videos. It redefines color fusion as a process that utilizes original full-frame textures and color editing information from non-edge areas to generate high-quality results.

**Strengths:**

1. The idea of MovingColor is very interesting and effective.
2. The structure of the article is clear and progresses logically.
3. The experiments conducted are relatively comprehensive.

**Limitations:**

1. This method appears to require a certain level of computational training cost. Some methods, such as "Ddcolor: Towards photorealistic image colorization via dual decoders [ICCV23]" and "Colorful image colorization [ECCV16]," retain the L channel values of the original video frames in the LAB space and replace the AB channels with the result's AB channel values. These methods can simply and effectively preserve structural edges, prevent color spillage, and require very short processing times. It is hoped that the authors will compare this post-processing approach and highlight the advantages of "MovingColor."

2. The applicability of this work is somewhat narrow. Although the concept is interesting, it heavily relies on the preliminary coloring results and the presence of some "obvious" boundaries (the cases chosen by the authors mostly have such boundaries).

3. The main text and supplementary materials of the paper and supplementary materials could include more cases, selecting a wider variety of scenes and environments. (Some do not clearly show the advantages, such as the green dog selected in the paper, which looks good in the picture but also has some artifacts in the video).

4. Add punctuation after formulas, line 274 in the paper.

**Suitability:**

3

---

### Official Review · Reviewer_j2LK · 2024-05-22

**Rating:** 5
**Confidence:** 3

**Summary:**

This paper introduces a novel video colour enhancement method, MovingColor,  that offers superior visual results by making precise adjustments to specific frame areas, avoiding flickering artifacts and unnatural colour blending at object boundaries due to coarse masks from current video segmentation algorithms. To achieve spatio-temporal inconsistencies, MovingColor utilizes a spectral-spatial hybrid encoder that captures multi-scale spatial and frequency features for better color adjustments in complex scenes. It also incorporates a global-local feature propagation module with Transformer blocks to ensure frame consistency.

**Strengths:**

1. This work firstly introduces a seamless colour fusion approach for videos, MovingColor, which tackles the problem of natural fusion of fine-grained video color adjustments.
2. This work builds the D5 dataset containing diverse 4K video clips, accurate semantic masks, ambient occlusion, reflection, sky mask, and depth, synthesised using 3D software. It can be used to comprehensively evaluate colour fusion performance and other video tasks.
3. The experiments, including various evaluation metrics, datasets, comparing methods, and  ablation study, are well designed and convincing.
4. The paper is well written with abundant supplementary materials.
5. The dataset and code will be released in the future.

**Limitations:**

1. Suboptimal Temporal Consistency: Although the method aims to achieve spatio-temporally consistent color fusion, it does not achieve state-of-the-art performance for several key temporal consistency metrics across three datasets. Meanwhile, it can be observed some noticeable flickering artifacts in provided demo videos. This suggests room for improvement in maintaining temporal coherence in the video sequences.
2. Clarification of Purpose and Application: The paper could benefit from a clearer explanation of the purpose and application scenarios for this task. It is not entirely clear whether this technique can be used for color fusion between two different videos or if it is intended solely for edge-aware color editing within local regions of a single video, functioning as a local region filter. Providing more discussion and examples would help to explain the potential use cases and benefits of this approach.

**Suitability:**

2

---

### Official Review · Reviewer_PmqP · 2024-05-24

**Rating:** 4
**Confidence:** 3

**Summary:**

This paper introduces MovingColor, a novel self-supervised training approach for fine-grained video color enhancement. This approach tackles flickering artifacts and difficulties of color blending at object boundaries due to unstable video segmentation. It incorporates full-frame textures and color for color fusion that takes into account spatiotemporal contexts to ensure consistency among frames. This proposed method demonstrates promising performance in terms of several evaluation metrics. However, additional clarification and comparisons may be needed for better understanding and evaluation.

**Strengths:**

1. D5 dataset is generated using D5 Render to address the lack of precise masks for evaluating video color enhancement.
2. A self-supervised training methodology is proposed for color fusion in video editing. A short video that vividly demonstrates this contributions is provided.
3. Multiple evaluation metrics are involved to evaluate the quality of video color fusion from different perspectives.

**Limitations:**

1. MovingColor is designed to “address spatio-temporal inconsistencies” according to the abstract and introduction, but the method only “performs competitively” in terms of temporal consistency compared with existing video editing methods according to Table 1 and Section 5.3. The framework seems to be similar to existing work such as video inpainting.
2. At the beginning of the second paragraph in the introduction section, “Harmonizer [14] and DeepLPF [24]” are mentioned as existing approaches for “video color enhancement”, but their main focus is to address image enhancement. The authors may want to improve the accuracy of description and cover comparisons between video color enhancement and related video-based techniques such as video inpainting and video harmonization.
3. What do the red and blue colors refer to in the third sub-figure of Figure 2 (a)?
4. In Section 5.2.2, MovingColor is mentioned as “the first to introduce a color fusion method”, but automated color fusion is also explored in ChromaFusionNet (AAAI 24) and other work. The authors may want to improve the accuracy of description and consider additional comparisons.

**Suitability:**

3

---

### Meta-Review · Area_Chair_6gWb · 2024-07-05

**Recommendation:** Accept (Oral)
**Confidence:** 4

**Metareview:**

The authors in this paper propose "MovingColor" for fine-grained color enhancement in videos through a self supervised training approach. The proposed method generates high quality results and promising performance based on several quantitative metrics.  The reviewers agree that there are several strengths - paper is well written, experimental results show good performance, dataset, and the propsed method is novel, etc. There are also some limitations in the paper that the reviewers elaborate below. Overall the reviewers agree that the paper can be accepted.  Please address the editorial and other issues that the reviewers identify in the camera ready version.